# Structural and functional analysis of hyper-thermostable ancestral L-amino acid oxidase that can convert Trp derivatives to D-forms by chemoenzymatic reaction

Yui Kawamura[1,5], Chiharu Ishida[1,5], Ryo Miyata[1,4], Azusa Miyata[1], Seiichiro Hayashi[2], Daisuke Fujinami[1], Sohei Ito [1] & Shogo Nakano [1,3✉]

Production of D-amino acids (D-AAs) on a large-scale enables to provide precursors of peptide therapeutics. In this study, we designed a novel L-amino acid oxidase, HTAncLAAO2, by ancestral sequence reconstruction, exhibiting high thermostability and long-term stability. The crystal structure of HTAncLAAO2 was determined at 2.2 Å by X-ray crystallography, revealing that the enzyme has an octameric form like a "ninja-star" feature. Enzymatic property analysis demonstrated that HTAncLAAO2 exhibits three-order larger $k_{cat}/K_m$ values towards four L-AAs (L-Phe, L-Leu, L-Met, and L-Ile) than that of L-Trp. Through screening the variants, we obtained the HTAncLAAO2(W220A) variant, which shows a > 6-fold increase in $k_{cat}$ value toward L-Trp compared to the original enzyme. This variant applies to synthesizing enantio-pure D-Trp derivatives from L- or rac-forms at a preparative scale. Given its excellent properties, HTAncLAAO2 would be a starting point for designing novel oxidases with high activity toward various amines and AAs.

[1] Graduate Division of Nutritional and Environmental Sciences, University of Shizuoka, 52-1 Yada, Suruga-ku, Shizuoka 422-8526, Japan. [2] Division of Structural Biology, Medical Institute of Bioregulation, Kyushu University, Higashi-ku, Fukuoka 812-8582, Japan. [3] PREST, Japan Science and Technology Agency, Saitama, Japan. [4] Present address: Health and Medical Research Institute, National Institute of Advanced Industrial Science and Technology, 2217-14 Hayashi-cho, Takamatsu, Kagawa 761-0395, Japan. [5] These authors contributed equally: Yui Kawamura, Chiharu Ishida. ✉email: snakano@u-shizuoka-ken.ac.jp

Amino acids (AAs) are fundamental biomolecules essential to all living organisms[1]. Hundreds of natural AAs, serving as precursors for many of bioactive molecules, including peptides, secondary metabolites, and pharmaceuticals, have been reported[2]. The α-AAs exist in two forms based on the chirality of the $C_\alpha$ atom: L-AAs and D-AAs, except for Gly[1]. L-AAs are broadly distributed in nature, with their physiological functions being extensively studied. For instance, twenty L-AAs, known as proteinogenic amino acids, are utilized as the building blocks of proteins[1,2]. Fermentation methods have been established to synthesize a large amount of L-AAs like L-Lys[3], L-Glu[4], and L-Phe[5], which find extensive applications. Conversely, the physiological functions of D-AAs remained elusive for a long time even though the compounds were widely observed in microbes, plants, animals, and foods[6–8]. Advances in detection technology have facilitated our understanding of their roles; D-Ser acts as a co-agonist of the *N*-methyl-D-aspartate receptor in the mammalian brain[9], and D-Asp functions as a neurotransmitter or neuromodulator[10]. Components of the bacterial cell wall, peptidoglycans, contain many D-AAs[8].

Given the physiological and chemical importance of D-AAs, it is crucial to develop methods for their large-scale synthesis to evaluate their functions[2]. Both chemical and enzymatic syntheses have been employed for this purpose[2,11–13], with this study focusing on the latter one. Several of enzymatic synthesis methods currently exist, such as the sequential conversion of hydantoin to D-amino acids via hydantoin racemization and D-selective degradation, which involve the use of three enzymes: hydantoin racemase, D-hydantoinase, and D-carbamoylase[14]. The reductive amination of keto acids into D-AAs is commonly adopted due to its capacity to generate D-AAs in large quantities with high enantio-purity[15]; the reaction can be performed by D-amino acid dehydrogenase[16], D-amino acid aminotransferase[17], and ω-transaminase[18]. L-amino acid deaminase (LAADs)[16,17,19–23] and L-amino acid oxidases (LAAOs)[15,24–27] are key biocatalysts efficiently converting L-AAs to keto acids. Both LAAOs and LAADs present distinct advantages and disadvantages for the synthesis. An advantage of LAAOs is that the enzyme only requires $O_2$ molecules to regenerate oxidized FAD. Thus, purified LAAOs can be utilized as biocatalysts. In contrast, LAADs necessitate electron carriers such as phenazine methosulfate or *E. coli* membranes for the regeneration[21–23]. However, LAAOs have a disadvantage in that they produce $H_2O_2$ as a by-product, which often leads to unexpected reactions during synthesis. In contrast, LAADs do not generate $H_2O_2$, thereby mitigating the risk of unexpected reactions[21]. Chemoenzymatic reactions utilizing these enzymes and chemical reductants as catalysts have been successfully employed for the deracemization or stereoinversion of racemic or L-AAs into D-forms[15]. The reaction efficiency could be improved with the availability of highly stable LAADs or LAAOs because the inactivation of the enzymes during the reaction would be suppressed. In a previous study, hyper-thermostable ancestral LAAOs (HTAncLAAOs) were designed by ancestral sequence reconstruction (ASR) using functionally uncharacterized FAD-dependent enzymes from *Chitinophaga* species as templates[27]. HTAncLAAO exhibited exceptional thermostability and long-term stability, making them suitable for keto acid synthesis from L-AAs[27]. However, the limited substrate selectivity and low specific activity of HTAncLAAO have been hurdles to its enzymatic application[27]. In addition, the relationship between the multimeric structure of HTAncLAAO and its high stability is of interest, but has not been resolved due to the difficulty of structural analysis. This also made challenging to design the variants of which the selectivity is broadened by structure-guided design.

In this study, we designed a HTAncLAAO (HTAncLAAO2) by ASR to obtain an accurate 3D structure of HTAncLAAO by X-ray crystallography. As well as other ancestral LAAOs[25–27], HTAncLAAO2 is also expressed by *E.coli* system as soluble form. HTAncLAAO2, which shared a high sequence identity (69.8%) with existing HTAncLAAO and can form high-quality crystals, was designed through ASR with additional screening. X-ray and electron microscopy (EM) revealed that HTAncLAAO2 exits in an octameric state. ASR, one of the sequence-based protein design methods, predicts progenitor sequences from the multiple sequence alignment and phylogenetic tree of descendant sequences; ancestral proteins often exhibited favorable properties for the application[28]. The enzymatic properties of ancestral enzymes, such as substrate selectivity, thermostability, and soluble expression level, are often enhanced[29–31]. The HTAncLAAO2 variant, which exhibits higher activity toward L-Trp, was designed using this structural information. Finally, enantio-pure D-Trp derivatives, which are expected to be precursors of indole-containing fine chemicals[17], were synthesized at a preparative scale using the designed variant through a chemoenzymatic reaction. D-Trp derivatives act as inhibitors of enteric pathogens[32] and play a role in bioactive secondary metabolites[33]. Establishing a reaction system to synthesize these derivatives can advance research in these areas by ensuring a consistent supply of the compounds.

## Results and discussion

**The overall structure of HTAncLAAO2.** The main mission of this study was to determine the crystal structure of HTAncLAAO. To accomplish this, we first executed a protein expression test and the screening of crystallization conditions for newly designed HTAncLAAOs created by ASR, modifying the combination of homologous sequences. High-quality crystals were finally obtained for only one protein, designated as HTAncLAAO2 (Table S2). HTAncLAAO2 was generated utilizing the four functionally uncharacterized FAD-dependent oxidoreductases from *Flavobacterium* species (Table S1). The soluble expression level of HTAncLAAO2 was reached to >20 mg/L (Table S3); the value was enough to apply the enzymes in the D-AA synthesis. HTAncLAAO2 bears a high sequence identity with the HTAncLAAO[27], suggesting that substrate recognition mechanisms would be conserved to each other. The overall structure of HTAncLAAO2, determined at a 2.2 Å resolution (Table S4), indicating that the enzyme has typical folding of amine oxidase superfamily, such as L-Lysine α-oxidase[34,35] and LAAO from *Pseudoalteromonas* species[26,36], in spite that they shared low sequence identity to each other. The unique point of HTAncLAAO2 is that the enzyme displays a distinctive octameric state (Fig. 1A). Here, it was crucial to ascertain whether this observed oligomer was a product of crystal packing or a natural state. We performed gel-filtration and negative-stained EM analyses to investigate the oligomeric state in physiological conditions. Gel-filtration chromatography analysis suggested that the HTAncLAAO2 eluted as 544 kDa (denoted by the red star in Fig. 1B), indicating that the formation of complexes in solution by eight HTAncLAAO2 molecules, each with a molecular weight of 63 kDa. Negative-stained EM analysis supports the ninja-star-like oligomeric structure, as indicated by X-ray crystallography, under physiological conditions (Fig. 1C).

The results show that HTAncLAAO2 has an octameric, ninja-star-like structure. This distinctive oligomeric state may be origin for the high thermostability and long-term stability of HTAncLAAOs; other thermostable enzymes also had an oligomeric form to improve the stability[37]. L-arginine oxidase, another class

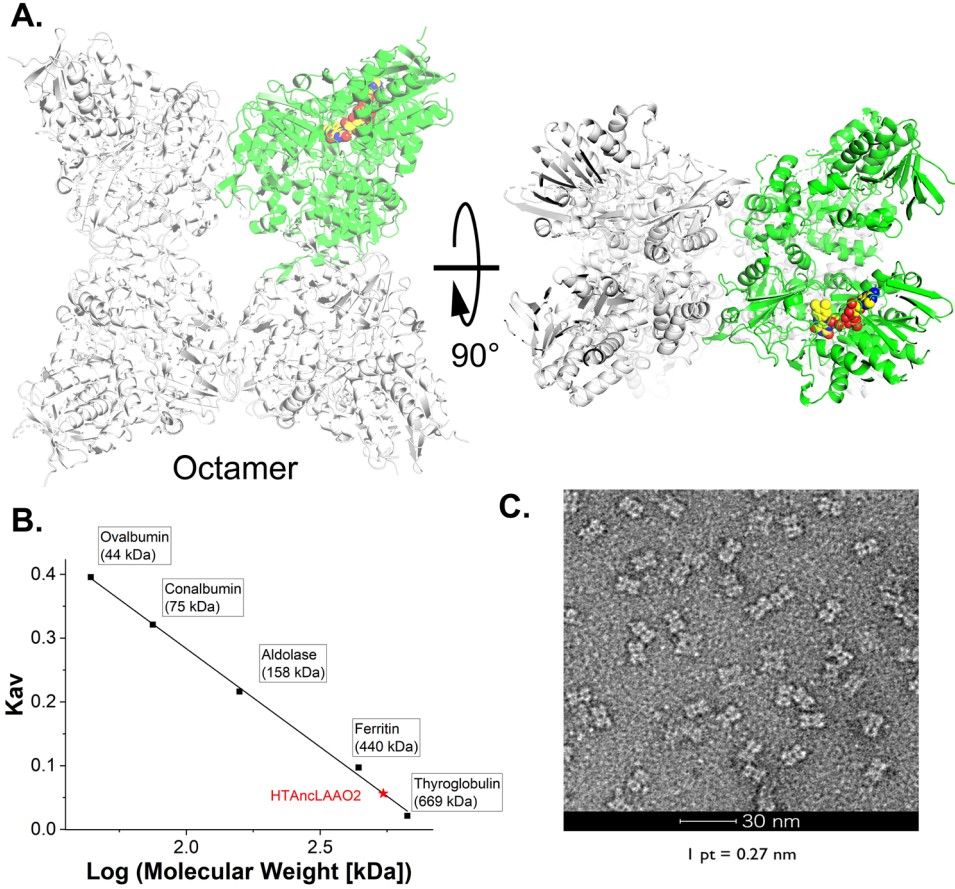

**Fig. 1 Structural analysis of HTAncLAAO2. A** As shown in the figure, the HTAncLAAO2 forms octamer in the crystal. **B** Size-exclusion chromatography analysis of HTAncLAAO2. According to the calibration curve, the molecular weight of HTAncLAAO2 was calculated as 544.4 kDa. The molecular weight of the monomer form of HTAncLAAO2 is 63 kDa. **C** Negative stain EM of HTAncLAAO2.

of thermostable enzymes with high specificity towards L-Arg[38,39], is also in an octameric state and showed high thermostability.

**Analysis of enzyme properties of HTAncLAAO2.** As described in the previous section, we successfully determined the crystal structure of HTAncLAAO2. The next mission was to validate that HTAncLAAO2 retained enzymatic activity compared with that of the HTAncLAAO[27]. Initial assessment of HTAncLAAO2's specific activity using 20 L-AAs as substrates revealed a high activity towards four L-AAs, including L-Phe, L-Met, L-Ile, and L-Leu, similar to the HTAncLAAO (Fig. 2A). Furthermore, weak activity was observed towards additional three L-AAs: L-Val, L-Tyr, and L-Trp (Fig. 2A). Enzyme kinetic parameters for HTAncLAAO2, estimated using five L-AAs, demonstrated $k_{cat}/K_m$ values an order of magnitude higher than those of HTAncLAAO (Table 1 and Fig. 2B). Subsequent analysis of the parameters for five L-AAs revealed apparent differences in the $k_{cat}/K_m$ values, while changes in the $k_{cat}$ values were relatively small (Table 1). The $k_{cat}/K_m$ values for the four hydrophobic L-AAs (L-Phe, L-Met, L-Leu, and L-Ile) were more than three orders of magnitude greater than that for L-Trp, but the $k_{cat}$ values were at most six times higher (Table 1). This suggests that the structure of HTAncLAAO2's active site and substrate entrance pathway was optimized for recognizing these four L-AAs, maximizing catalytic efficiency. Alterations in the structure to better recognize other L-AAs could potentially enhance HTAncLAAO2's activity towards substrates that currently show weak activity. The thermostability (red colored in Fig. 2C) and long-term stability (red colored in Fig. 2D) of HTAncLAAO2, measured using L-Met as a substrate,

were similar to those of HTAncLAAO; a $T_{1/2}$ of about 85 °C, and residual activity exceeding 95% following a 7-day incubation at 30 °C (Fig. 2C, D). This suggests that HTAncLAAO2, like the HTAncLAAO, possesses suitable properties to conduct D-AA synthesis.

**NOS-bridge between the side chain of K304 and C505 in HTAncLAAO2.** We unexpectedly found a NOS bridge between residues K304 and C505 which are located remotely from the active site (Fig. 3A). In many enzymes, the formation of such a bridge influences activity[40,41]. We thus investigated whether this bridge in HTAncLAAO2 is essential for its activity. Enzyme kinetic parameters of HTAncLAAO2 were determined for the samples in the presence and absence of 1 mM TCEP (Fig. 3B). The results indicated that the presence of reductants had a negligible impact on these parameters. Specifically, for native HTAncLAAO2, $k_{cat}$ and $k_{cat}/K_m$ values decreased by about 10% and 40 for samples containing TCEP, respectively (Table 2 and Fig. 3B–D). A similar level of parameter decrease was observed in variants K304A and C505A (Table 2), which were designed to cleave the NOS bridge in HTAncLAAO2. HTAncLAAO2 and these variants exhibited virtually identical thermostability ($T_m$ values ranging from 82.0 to 82.9 °C, as shown in Table 2), specific activity toward 20 L-AAs (Table S5), and enzyme kinetic parameters (Table 2, Fig. 3E). Gel-filtration chromatography analysis of the variants indicated that the NOS bridge is not affected to form the oligomer state of HTAncLAAO2; the variants had 8-mer in solution condition as well as the native form (Fig. S2).

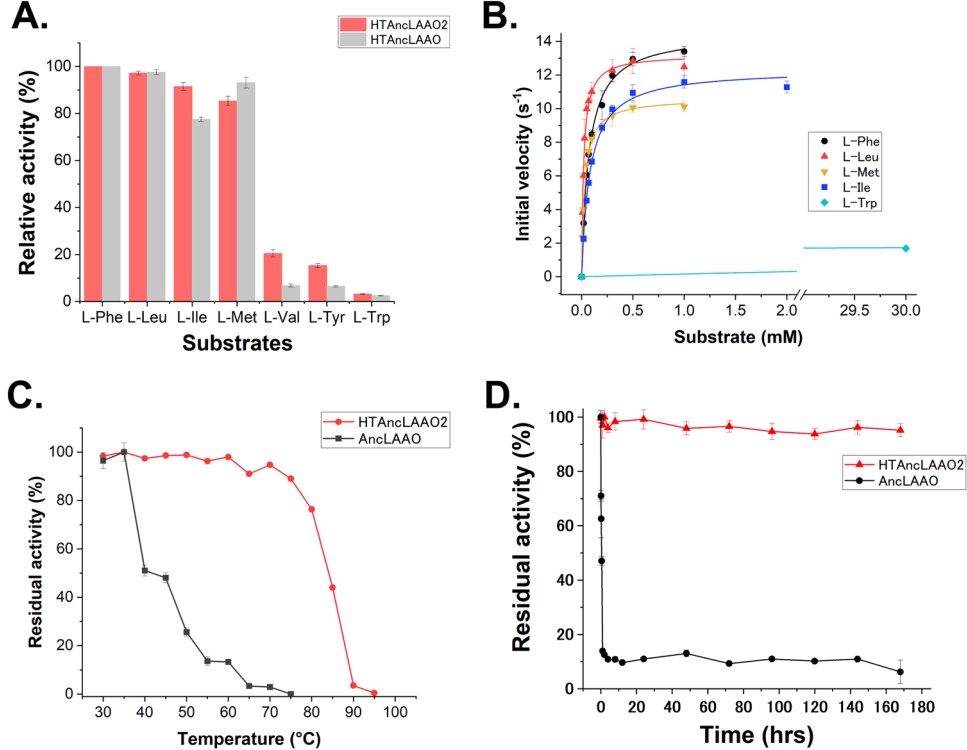

**Fig. 2 Enzyme functional analysis of HTAncLAAO2. A** The specific activity of HTAncLAAOs toward seven L-amino acids. The relative activity was calculated by normalizing the parameter of L-Phe to 100%. The activity for HTAncLAAO and HTAncLAAO2 was represented as gray and red bar, respectively. **B** Enzyme kinetic plots of HTAncLAAO2. The initial velocity of HTAncLAAO2 toward L-Phe, L-Leu, L-Met, L-Ile, and L-Trp are shown as a black circle, red upward triangle, orange downward triangle, blue square, and light blue diamond, respectively. The parameters are shown in Table 1. The data are shown as the mean ± SD. **C**, **D** Temperature-dependent activity change and long-term thermostability of HTAncLAAO2. Residual activity was calculated by normalizing each sample's highest activity to 100%. The activities of HTAncLAAO2 and AncLAAO are plotted as red circles and black squares, respectively.

**Table 1 Enzyme kinetic parameters of HTAncLAAO2 toward five L-amino acids[a].**

| Substrate | HTAncLAAO2 | | | HTAncLAAO[b] | | |
| | $k_{cat}$ $s^{-1}$ | $K_m$ mM | $k_{cat}/K_m$ $s^{-1}$ $mM^{-1}$ | $k_{cat}$ $s^{-1}$ | $K_m$ mM | $k_{cat}/K_m$ $s^{-1}$ $mM^{-1}$ |
| --- | --- | --- | --- | --- | --- | --- |
| L-Phe | 14.5 ± 0.2 | 0.071 ± 0.004 | 204 | 8.7 ± 0.1 | 0.3 ± 0.0 | 29.0 |
| L-Leu | 13.2 ± 0.3 | 0.020 ± 0.002 | 661 | 8.9 ± 0.2 | 0.1 ± 0.0 | 89.0 |
| L-Met | 10.6 ± 0.1 | 0.031 ± 0.001 | 343 | 8.7 ± 0.1 | 0.2 ± 0.0 | 43.5 |
| L-Ile | 12.3 ± 0.0 | 0.081 ± 0.005 | 152 | 5.1 ± 0.1 | 0.9 ± 0.0 | 5.7 |
| L-Trp | 2.6 ± 0.1 | 15.4 ± 1.1 | 0.17 | n.d. | n.d. | n.d. |

[a]The measurement of enzyme kinetic parameters was performed independently three times ($N = 3$).
[b]The enzyme kinetic parameters of HTAncLAAO were cited from the following ref. [27].

In conclusion, the NOS bridge identified in HTAncLAAO2 had minimal influence on in vitro parameters such as enzyme activity and thermostability. HTAncLAAO2 generates $H_2O_2$ during the reaction, potentially creating NOS bridge between K304 and C505, which are located in close proximity. These findings indicate that the NOS bridge does not always influence enzyme activity, suggesting the necessity for experimental analysis to evaluate the properties of this bridge. There is also possibility that the bridge might have other functions that cannot be characterized by in vitro assays.

**Structure-guided mutagenesis of HTAncLAAO2 to improve activity toward L-Trp.** Structural and functional analysis indicated that HTAncLAAO2 exhibited almost identical enzymatic properties to the HTAncLAAO[27]. The next challenge is to design

a variant of HTAncLAAO2 with broader substrate selectivity, particularly towards L-AAs that are not preferential substrates for HTAncLAAO2 (Fig. 2A). To this end, we attempted to create an HTAncLAAO2 variant with high activity towards L-Trp.

To understand the substrate recognition mechanism, we first predicted the substrate accessibility by CAVER software[42], comprised of many hydrophobic residues, four of which (W220, F225, Y383, and W520) are aromatic residues (Fig. 4A). In Fig. 4, we represented the active site structure of HTAncLAAO2 by superimposing the L-Trp binding form of AncLAAO-N5[26] based on the FAD molecule. This configuration potentially facilitates introducing hydrophobic L-AAs into the active site more efficiently. The functional importance of other residues could be predicted from the structural and functional analysis of other LAAOs (Fig. S3). For instance, R67 and K322 are

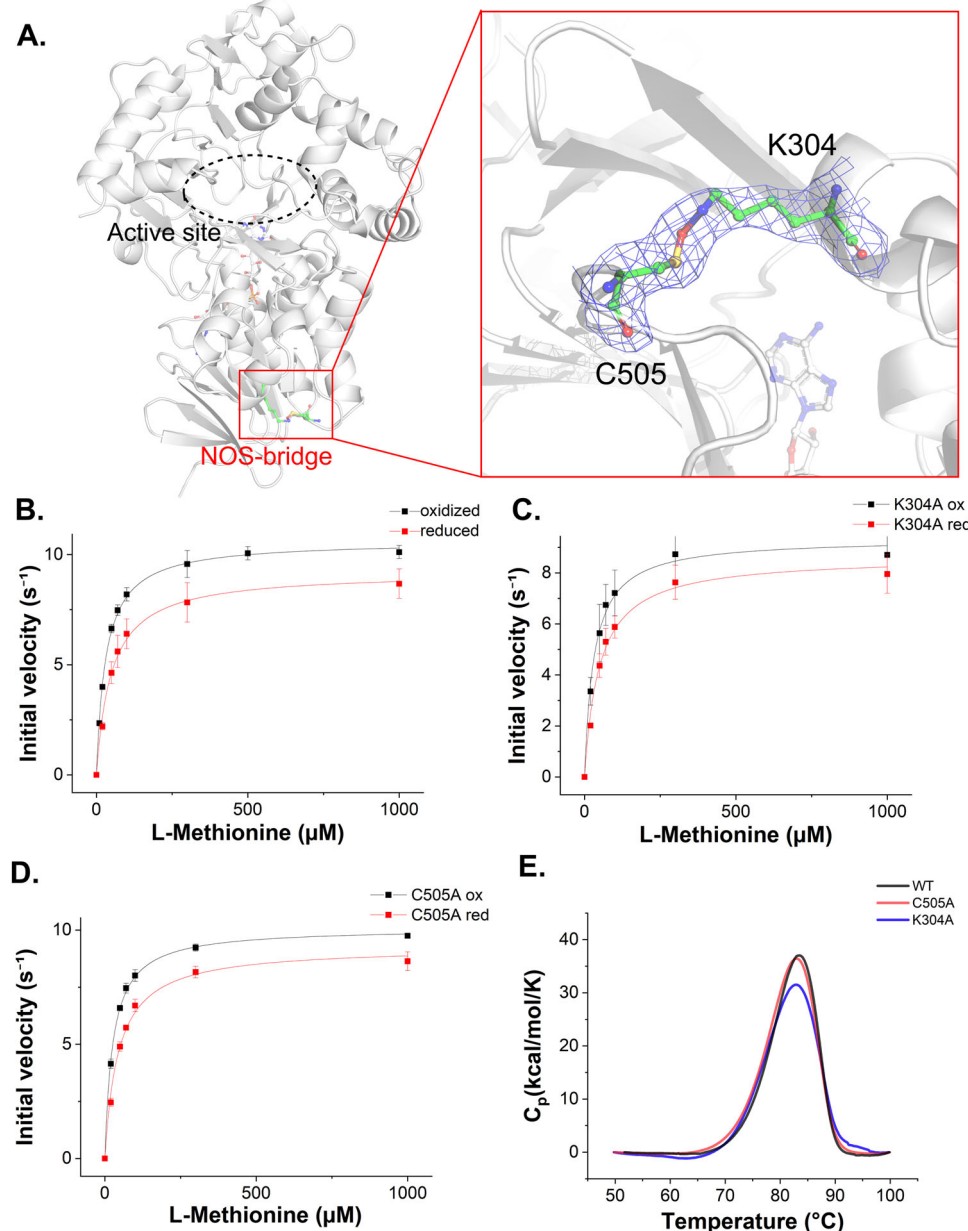

**Fig. 3 Structural and functional analysis of NOS-bridge in the HTAncLAAO2. A** Electron density map of the NOS-bridge. As shown in the 2F$_o$-F$_c$ electron density map, which is contoured at 1.0 σ, K304, and C505 formed the bridge. **B–D** Enzyme kinetic plots of HTAncLAAO2 (**B**), K304A (**C**), and C505A variants (**D**) toward L-Met. The initial velocities were measured under the conditions with (red square) or without (black circle) containing chemical reductants (TCEP). The data are shown as the mean ± SD. **E** DSC isotherms of HTAncLAAO2 and their variants. The isotherms for HTAncLAAO2, K304A and C505A were plotted by black, blue, and red lines, respectively.

residues that recognize the carboxyl group of the substrate and O$_2$, which oxidizes the reduced FAD post-reaction, respectively (Fig. 4A).

Despite numerous attempts, the L-Trp binding structure of HTAncLAAO2 could not be determined because no crystals providing sufficient resolution were obtained. Alternatively, we predicted L-Trp binding form of HTAncLAAO2 by superposing the L-Trp binding structure of AncLAAO-N4 based on FAD molecule (Fig. 4B). The HTAncLAAO2 structure suggested that the side chains of five hydrophobic residues (W220, Y383, W482, and W520 depicted in Fig. 4B, and L320 in Fig. S3) were located proximal to the indole group of L-Trp. HTAncLAAO2 could enhance its activity towards L-Trp by eliminating the side chain groups of residues potentially interacting with the substrate.

HTAncLAAO2 variants, with these five residues mutated to Ala, were designed to validate this hypothesis. We managed to purify three of these variants: L320A, Y383A, and W220A. The other two variants were expressed as insoluble fractions, and we could not purify samples for further enzymatic property estimation. Enzyme kinetic parameters of these HTAncLAAO2 variants were estimated using five L-AAs (L-Phe, L-Leu, L-Met, L-Ile, and L-Trp) as substrates. The results suggested that $k_{cat}/K_m$ values towards the four L-AAs, excluding L-Trp, decreased by the mutations (Table S6 and Fig. 4C). However, the $k_{cat}/K_m$ value towards L-Trp increased more than 10-fold for W220A compared to the template HTAncLAAO2 (Table S6). Enzyme kinetic plots of W220A (Fig. 4D) suggest that this improvement in $k_{cat}/K_m$ was mainly contributed by the enhancement of $k_{cat}$ value, which was

16.0/sec for W220A compared to 2.6/sec for template HTAn-cLAAO2. The relative activity of the HTAncLAAO2(W220A) variant towards 20 L-AAs was evaluated, showing a selectivity conversion because of the mutation; this variant acquired the

activity toward long-chain L-AAs (L-Trp, L-Lys and L-Arg) by sacrificing with the activity toward L-Ile and L-Val (Table S5). Enzyme kinetic analysis suggested that Tte W220A mutation likely introduces a cavity in the active site, accommodating the long-chain L-AAs while compromising affinity for L-Phe, L-Leu, L-Met, and L-Ile. Indeed, the Km values of HTAncLAAO2(native) for these four L-AAs were at least 3-fold lower than those of the HTAncLAAO2(W220A) variant (Table S6).

In summary, we successfully designed an HTAncLAAO2 variant with high activity towards L-Trp, namely the W220A variant. The enhancement of the $k_{cat}$ value through mutation suggests that removing the side chain of the indole group from W220 allows HTAncLAAO2 to recognize L-Trp more favorably.

**Table 2 Enzyme kinetic parameters of HTAncLAAO2 and their variants (K304A and C505A) to predict the function of NOS binding[a].**

| Sample | TCEP | $k_{cat}$ s$^{-1}$ | $K_m$ μM | $k_{cat}/K_m$ s$^{-1}$ μM$^{-1}$ | $T_m$[b] °C |
|---|---|---|---|---|---|
| native | | | | | |
| | − (ox) | 10.6 ± 0.1 | 31.4 ± 1.3 | 0.34 | 84.0 ± 0.3 |
| | + (red) | 9.2 ± 0.2 | 49.5 ± 4.5 | 0.19 | |
| K304A | | | | | |
| | − (ox) | 9.4 ± 0.2 | 31.4 ± 3.3 | 0.30 | 83.3 ± 0.2 |
| | + (red) | 8.6 ± 0.3 | 49.4 ± 5.5 | 0.18 | |
| C505A | | | | | |
| | − (ox) | 10.1 ± 0.1 | 26.8 ± 1.1 | 0.38 | 83.1 ± 0.1 |
| | + (red) | 9.3 ± 0.2 | 44.7 ± 4.1 | 0.21 | |

[a]The measurement of enzyme kinetic parameters was performed independently three times ($N = 3$).
[b]The $T_m$ values were estimated utilizing samples without containing TCEP because the reductants affected to the thermodiagrams of DSC.

**Conversion of Trp derivatives to D-forms utilizing HTAn-cLAAO2(W220A) variant.** Considering the broad substrate specificity and high thermostability, W220A is helpful for application in the stereoinversion or deracemization of Trp derivatives to D-forms at a preparative scale due to its high activity and stability. To validate this, we initially attempted the stereoinversion of L-Trp using the scheme depicted in Fig. 5A. In this schema, L-Trp was oxidized to imino acids by HTAn-cLAAO2(W220A), and the products were concurrently reduced to D, L-Trp by chemical reductants, NH$_3$:BH$_3$ (Fig. 5A). Catalase converted H$_2$O$_2$, a byproduct of the reaction, to H$_2$O and O$_2$. Under conditions containing 204 mg of L-Trp, the time-dependent conversion of L- to D-forms was assessed with HPLC analysis. The reaction was performed under varying temperature conditions, ranging from 30 to 50 °C (Fig. 5B),

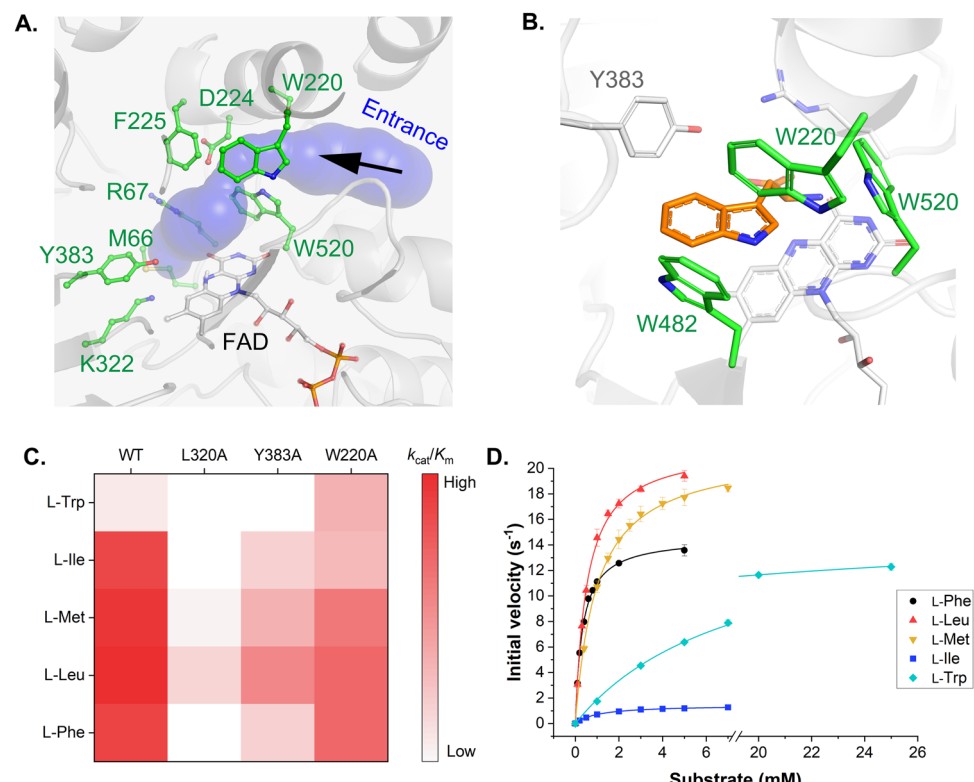

**Fig. 4 Screening of HTAncLAAO2 variants that exhibits high activity toward L-Trp derivatives. A** Substrate entrance pathway of HTAncLAAO2. The residues surrounding the pathway were shown as green stick and ball models. **B** Active site structure of HTAncLAAO2. L-Trp (orange) was docked to HTAncLAAO2, referring to the structure of the L-Trp binding form of AncLAAO-N5. **C** Differences of $k_{cat}/K_m$ values of HTAncLAAO2 and variants indicated by the color gradient. In the figure, we showed the parameters of HTAncLAAO2 (WT) and their variants (L320A, Y383A, and W220A). The activity was measured for five L-amino acids (L-Trp, L-Ile, L-Met, L-Leu, and L-Phe). **D** Enzyme kinetic plots of HTAncLAAO2(W220A) variant toward the five L-amino acids. The initial velocities, which were obtained by utilizing L-Phe, L-Leu, L-Met, L-Ile, and L-Trp as substrates, were colored black, red, orange, blue, and light blue, respectively. The data are shown as the mean ± SD.

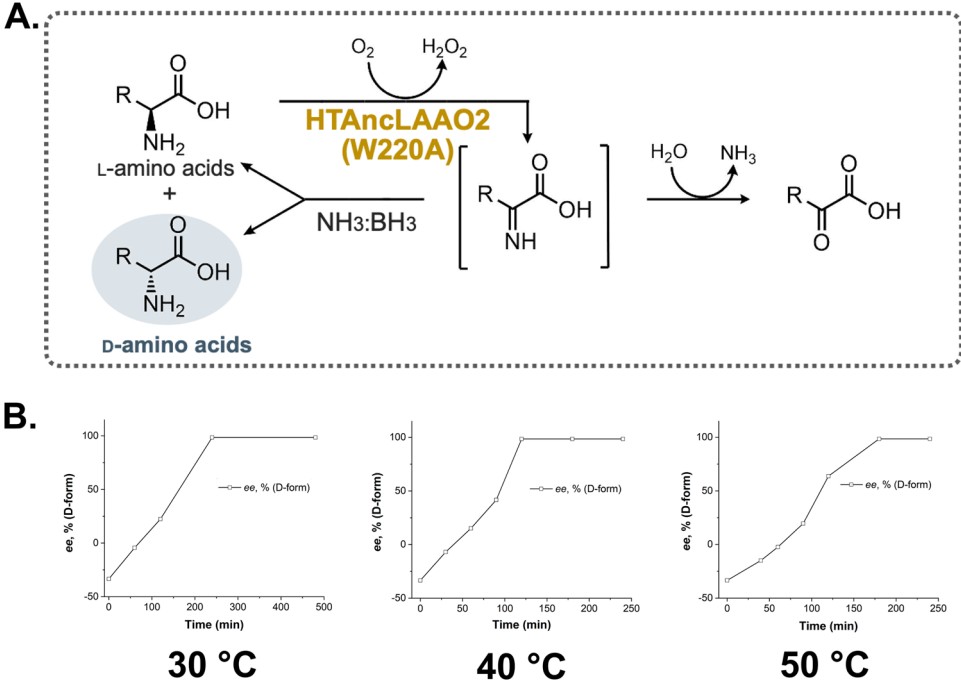

**Fig. 5 Optimization of reaction condition to synthesize D-Trp utilizing HTAncLAAO(W220A) variant. A** Reaction cascade to obtain enantio-pure D-amino acids from L-forms utilizing HTAncLAAO2. **B** Time course for production of D-Trp from L-Trp by changing the reaction temperature. The stereoinversion of L-Trp to D-form was achieved in every measured temperature (30, 40, and 50 °C). At 40 °C condition, the reaction was most efficiently progressed.

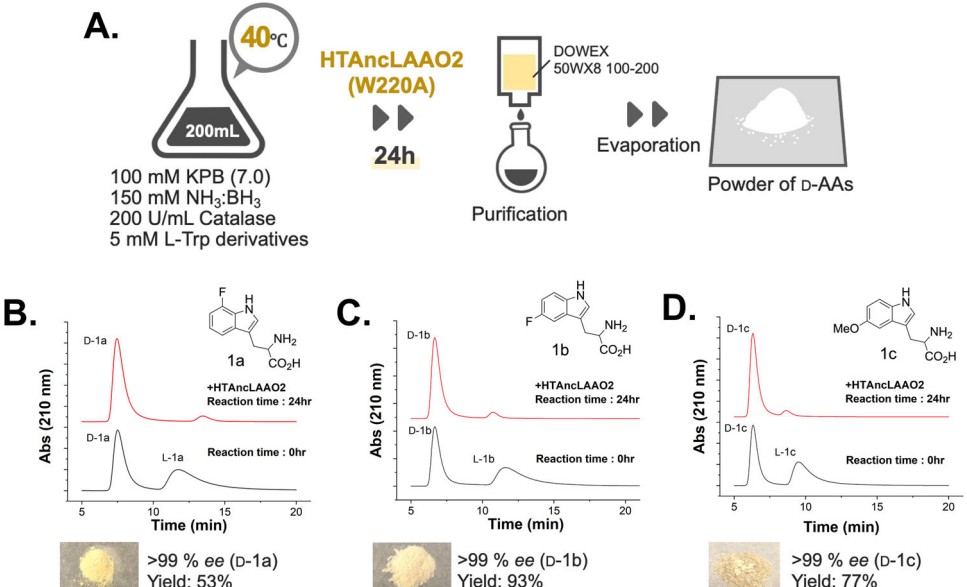

**Fig. 6 Preparative scale synthesis of D-Trp derivatives from their racemates utilizing HTAncLAAO2(W220A). A** Schematic view to synthesize and purify D-Trp derivatives from racemates at preparative scale. The detailed purification procedures were written in the Material and Method section. **B–D** HPLC analysis of reaction solution before and after the deracemization (1a for **B**, 1b for **C**, and 1c for **D**). The *ee* value and the yields of the D-forms were represented in the figure. The products were characterized by HRMS.

resulting in *ee* values exceeding 99% in all situations (Fig. 5B). Differences were observed in reaction efficiency, with L-Trp fully converted to D-forms within 150 min at 40 °C, as opposed to approximately 250 min at 30 °C and 180 min at 50 °C, respectively (Fig. 5B). Consequently, we selected 40 °C as the reaction temperature for subsequent conditions.

Next, we attempted the stereoinversion and deracemization of Trp derivatives at a preparative scale, employing the reaction and

purification conditions depicted in Fig. 6A. The reaction was conducted with 200 mL of the reaction solution at 40 °C. The synthesized D-Trp derivatives were purified by ion exchange chromatography. Powders were obtained after solvent evaporation (Fig. 6A). Utilizing the synthetic protocols illustrated in Fig. 6A, the reaction was performed using 7-fluoro-D,L-Trp (1a in Fig. 6B), 5-fluoro-D,L-Trp (1b in Fig. 6C), and 5-methyl-D,L-Trp (1c in Fig. 6D) as substrates. All Trp derivatives were fully

converted to D-forms after a 24-hour reaction (from black to the red line in Fig. 6B–D). The yields of the purified D-Trp derivatives exceeded 53% with high *ee* values (>99%). The D-Trp derivatives were characterized with HRMS analysis (Fig. S4).

In conclusion, we have demonstrated that chemoenzymatic reactions can apply the HTAncLAAO2(W220A) variant to stereoinversion and deracemization of Trp derivatives to D-forms. The results suggest that the HTAncLAAO2(W220A) variant maintains its high stability compared to the HTAncLAAO.

## Conclusion
In the present study, we successfully determined the crystal structure of HTAncLAAO2, which was redesigned from existing databases to produce high-quality crystals. The unique octamer form of the HTAncLAAO2 structure, similar to a 'ninja-star' feature, may enhance the thermostability of HTAncLAAOs, similar to cases observed with other highly oligomerized enzymes commonly found in thermophiles[37]. HTAncLAAO2 demonstrated substrate selectivity almost identical to, and higher activity than, the HTAncLAAO[27]; these properties make the enzyme suitable for synthesizing D-AAs from L- or *rac*-AAs. We successfully synthesized enantiopure D-Trp derivatives at a preparative scale through a chemoenzymatic reaction, using the HTAncLAAO2(W220A) variant as a biocatalyst. The HTAncLAAO2(W220A) was screened through the functional analysis of their variants, designed through Ala scanning of substrate recognition residues. The structural information of HTAncLAAO2 will prove invaluable in designing variants whose substrate selectivity can be altered towards other compounds. This can be achieved by combining colorimetric detection methods with site-directed mutagenesis analysis of the active site residues. We propose that HTAncLAAO2 presents a promising candidate for the design of new oxidases with high activity towards various amines or amino acids, similar to D-amino acid oxidases[43,44] and monoamine oxidases[45,46].

## Materials and methods
**Screening and reconstruction of HTAncLAAO2**. In our study, using the screening procedure detailed in Fig. S1, we identified four sequences annotated as FAD-dependent oxidoreductases. The following two data are utilized as inputs: a template annotating a hypothetical protein from *Caulobacter radius* (CrHyp) and a library of 228 homolog sequences. Through the selection of sequences in the library where the 3rd, 213th, 355th, 398th, 514th, and 564th residues were conserved as Ile, Ser, Thr, Ile, Ser, and Phe respectively, we obtained the ISTISF library (Table S1), containing the four sequences as mentioned above. These six key residues, predicted as correlated residues, served as a motif for classification as well as the previous study[47]. Employing a custom Python script, "MAFFT2ASR.py" (available at https://github.com/shognakano/MAFFT2ASR), we designed an ancestral sequence named HTAncLAAO2 using the four sequences as inputs. This script offers a pipeline to execute the following tasks: 1. Multiple sequence alignment by MAFFT[48], 2. Phylogenetic tree generation through PhyML[49], and 3. Construction of an ancestral sequence using PAML[50]. The JTT empirical models were adopted for the analysis.

**Overexpression and purification of HTAncLAAO2 and their variants**. A DNA sequence encoding HTAncLAAO2 (see Table S2) was obtained by gene synthesis services from GeneScript. The purchased DNA fragments encoding HTAncLAAO2 were subsequently subcloned into the pET28a vector, prepared

through digestion by *Nco*I and *Xho*I. The resulting expression plasmids were transformed into *E. coli* strain BL21(DE3). These cells were propagated in 1 L of LB broth supplemented with 30 μg/mL kanamycin at 37 °C. When the O.D.600 value reached a range of 0.6–0.8, the temperature was reduced to 18 °C, and isopropyl-β-D-thiogalactopyranoside was added into the culture at a final concentration of 0.5 mM. After 16 h of cultivation, cells were collected by centrifugation. The cells were then resuspended in BufferA (20 mM potassium phosphate [pH 7.0] and 10 mM NaCl) and sonicated. The supernatant was collected by centrifugation at 11,000 g for 30 min and loaded onto a HisTrap-HP column (GE Healthcare). After washing the column with 30 mL of BufferA with 40 mM imidazole, the samples were eluted using 30 mL of BufferA with 100 mM imidazole. The protein expression level of HTAncLAAO2 was estimated by measuring the concentration of the samples after purification by Ni-affinity chromatography (Table S3). The samples were then loaded to a MonoQ 4.6/100 PE column (GE Healthcare), equilibrated with BufferA, and eluted with linear gradient method by mixing BufferA and BufferB (20 mM potassium phosphate [pH 7.0] and 400 mM NaCl). Fractions containing the samples were concentrated and loaded onto a Superdex 200 Increase column (GE Healthcare) equilibrated with BufferA. The purity of the samples was checked by SDS-PAGE. The concentration of HTAncLAAO2 was estimated by measuring absorbance at 280 nm, with the molar extinction coefficient at 280 nm calculated with Protein Calculator v3.4. These purified samples were utilized in subsequent experiments. The purification procedure for HTAncLAAO2 was conducted following the same process.

**Crystallization and structure determination of HTAncLAAO2**. The purified HTAncLAAO2 samples were concentrated to about 20 mg/mL by centrifugal filter unit, Amicon ultra-15 centrifugal filter 10 kDa MWCO (Millipore). HTAncLAAO2 crystals were obtained at 4 °C by mixing 1.5 L of the concentrated samples with the 1.0 L of the reservoir solution. The contents of the reservoir solution are 10% PEG3350, 0.2 M ammonium citrate tribasic, and 4%(v/v) 1,1,1,3,3,3-Hexafluoro-2-propanol. The crystals were quickly soaked into a cryo-protectant, the reservoir solution containing 20%(v/v) glycerol, and flash-cooled under a liquid nitrogen stream (100 K). X-ray diffraction data were collected at the BL5A beamline in the Photon Factory (Tsukuba, Japan). Integration and scaling were performed by XDS[51] and SCALA[52], respectively. The initial phase determination was achieved by molecular replacement method by PHASER[53]; here, the model structure of HTAncLAAO2 generated by AlphaFold2[54] was utilized as a template. Model building and refinement were performed by COOT[55] and either REFMAC[56] or PHENIX[53], respectively. All figures were prepared by PyMOL[57]. Crystallographic parameters are shown in Table S4.

**Electron microscopy**. A 3 μl of the purified HTAncLAAO2 (0.5 mg/ml) was applied to a copper grid supporting a continuous thin-carbon film, left for 1 min, and then stained with three drops of on-ice-cooled 2% uranyl acetate. Images of molecules were recorded by an Eagle 2k CCD camera (FEI, Hillsboro, OR) using a Tecnai T20 electron microscope (FEI) operated at an accelerating voltage of 200 kV, at a nominal magnification of ×80,000.

**Site-directed mutagenesis of HTAncLAAO2**. Methylated plasmids were employed as the template for site-directed mutagenesis, prepared by inserting the DNA sequence encoding HTAncLAAO2 into a pET28a vector. The mutation was introduced using the QuikChange Lightening Multisite Mutagenesis kit (Agilent technology). Primers utilized for variant design are

listed in Table S8. DNA sequencing was employed to confirm the sequences of the HTAncLAAO2 variants. The production and purification of the variants were performed by adopting the same procedures as those used for HTAncLAAO2. Here, the FAD content of HTAncLAAO2 and its variants was estimated following a similar procedure reported in previous studies[26,27,34]. The FAD contents of HTAncLAAO2 and their variants ranged from 57 to 60%, suggesting that the difference in content is unlikely to impact the relative comparison of $k_{cat}$ and $k_{cat}/K_m$ values.

**Analysis of enzymatic properties of HTAncLAAO2 and their variants.** The enzymatic activity of HTAncLAAO2 and its variants was assessed by quantifying the concentration of $H_2O_2$ by a colorimetric method. With reference to the previous study, the assay was performed utilizing the reaction buffer as follows: 100 mM bis-tris-HCl (pH 7.0), 1.5 mM 4-aminoantipyrine, 2.0 mM N-Ethyl-N-(2-hydroxy-3-sulfopropyl)-3-methylaniline, and 50 U/mL horse radish peroxidase. The substrate specificity was evaluated using the reaction buffer with 10 mM L-AAs. The thermostability and long-term stability were assessed using the reaction buffer containing 10 mM L-Met. Before initiating the measurement, diluted HTAncLAAO2 samples were incubated for 10 min at 30 to 95 °C to evaluate thermostability. Long-term stability was estimated by monitoring the activity of diluted samples incubated for 0–7 days at 30 °C. The initial velocity of HTAncLAAO2 was calculated by monitoring time-dependent absorption changes at 555 nm with a UV-Vis spectrometer. The molar extinction coefficient of the produced pigment at 555 nm was 39,200 $M^{-1} \cdot cm^{-1}$. The steady-state kinetic parameters of HTAncLAAO2 and its variants were estimated using five L-AA substrates (L-Phe, L-Leu, L-Met, L-Ile, and L-Trp). Enzyme kinetic parameters were estimated by fitting the initial velocity to the Michaelis-Menten equation using the non-linear least square method with ORIGIN software. These parameters are presented in Table 1 and Table S6.

**Deracemization and stereoinversion of rac- and L-Trp derivatives to D-forms.** A 200 mL of reaction solution containing 100 mM potassium phosphate (pH 8.0), 150 mM $NH_3:BH_3$, 40 kU catalase, and 5 mM of substrates L-Trp, L-**1a**, D,L-**1b**, and D,L-**1c** was prepared. The reaction was initiated by adding 0.4 mg of HTAncLAAO2(W220A). The time course of stereoinversion from L-Trp to D-Trp was monitored to determine the optimal reaction temperature. Samples of 100 µL from the reaction solution were collected after incubation at 30, 40, and 50 °C for 0–8 h. These samples were immediately mixed with 900 µL of a 1.15% (w/v) $HClO_4$ solution to stop the reaction, and the supernatant was subjected to HPLC analysis.

Deracemization (for D,L-**1a**, D,L-**1b** and D,L-**1c**) was conducted following the same procedure used for the stereoinversion of L-Trp, with the reaction temperature set to 40 °C. Reaction solutions were collected before (0 h) and after (24 h) the conversion for HPLC analysis. The enantiomeric excess (ee) values were estimated from the peak areas of the D- and L-forms. The synthesized D-**1a** to **1c** was purified as per the procedures reported by Parmeggiani et al.[16], and the isolated yields were calculated by quantifying the weights of the resultant powders.

**HPLC analysis.** The progression of the deracemization reaction was analyzed using reverse-phase high-performance liquid chromatography (HPLC). The liquid chromatography system, Prominence (Shimadzu), was utilized for this purpose, equipped with a UV-Vis detector (SPD-20AV, Shimadzu) and a CROWNPACK CR-I(+) column (length/internal diameter = 150/3.0 mm; DAICEL, Osaka, Japan). Absorption spectrum changes at 210 nm were monitored to detect reaction products. Detailed measurement conditions, including retention time, flow rate, oven temperature, and mobile phase composition, are presented in Table S7. The enantiomeric excess [ee (%)] was calculated using the following equation:

$$ee(\%) = \left[ (D_{area}) - (L_{area}) \right] / \left[ (D_{area}) + (L_{area}) \right] \times 100$$

Here, $D_{area}$ and $L_{area}$ represent the peak area of HPLC corresponding to D- or L-isomers, respectively. Chemical assignment of the products was performed by HRMS analysis (Q Exactive (Thermo Fisher Scientific, MA)).

**Reporting summary.** Further information on research design is available in the Nature Portfolio Reporting Summary linked to this article.

## Data availability
Crystal structure data of HTAncLAAO2 that support the findings of this study have been deposited in RCSB PDB with the following accession code: PDB ID: 8JHE. The validation report of 8JHE is available from Supplementary Data 1.

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

## Acknowledgements

This work was supported by JSPS KAKENHI Grant Numbers 18K14391 and 21K05395 and by JST, PRESTO Grant Number JPMJPR20AB. The authors thank Dr. D. Kohda for technical assistance with the electron microscope. This work was partly performed in the Cooperative Research Project Program of the Medical Institute of Bioregulation, Kyushu University.

## Author contributions

Y.K. and C.I. performed the enzyme kinetics assay. S.N. performed X-ray crystallography. R.M., A.M. performed an HRMS analysis of the obtained compounds. S.H. and D.F. performed the analysis of negative stained electron microscopy. S.N. managed this study. S.N. and S.I. contributed to the writing of the manuscript.

## Competing interests

The authors declare no competing interests.
