## [Peer Review File · Communications Chemistry]

Reviewers' comments:

Reviewer #1 (Remarks to the Author):

The crystal structure of HTAncLAAO2 was determined at 2.2 Å by X-ray crystallography, which exhibiting high thermostability and long-term stability. Enzymatic property analysis were demonstrated and this variant was applied to synthesizing enantio-pure D-Trp derivatives from L- or rac-forms at a preparative scale. The results are important and attractive. There are some points should be addressed before publication.

1. It is claimed that initial assessment of HTAncLAAO2's specific activity is similar to the HTAncLAAO (Fig. 2A). The specific activity of HTAncLAAO should be given in Fig 2A.
2. The NOS bridge identified in HTAncLAAO2 had minimal influence on in vitro parameters such as enzyme activity and thermostability. It is unimportant and should be simplified.
3. The authors attempted to create an HTAncLAAO2 variant with high activity towards L-Trp. Why L-Trp? How about other L-AAs that are not preferential substrates for HTAncLAAO2? At least it should be discussed. If more L-AAs were catalyzed by other HTAncLAAO2 variants guided by structural information, the manuscript will be more important.

Reviewer #2 (Remarks to the Author):

In this study, the authors designed a novel L-amino acid oxidase, HTAncLAAO2, by ancestral sequence reconstruction, exhibiting high thermostability and long-term stability. Through screening the variants, they obtained the HTAncLAAO2(W220A) variant, which shows >6-fold increase in kcat value toward L-Trp compared to the original enzyme. This variant applies to synthesizing enantio-pure D-Trp derivatives from L- or rac-forms at a preparative scale. Given its excellent properties, HTAncLAAO2 would be a starting point for designing novel oxidases with high activity toward various amines and AAs. This is an interesting work and is promising to be published in this journal after addressing the following concerns.

1. The activity detection of the control group (AncLAAO) in Fig.2D at 30°C should be set at a smaller time interval. From the current experimental results, it cannot be seen that the activity of AncLAAO decreases over time.
2. Reported studies showed that NOS-bridge can affect the structure, function, and stability of proteins. However, the results of this study show that NOS-bridge only has a certain effect on the activity of HTAncLAAO2, but has no significant effect on its thermal stability. What is the reason? In my opinion, It is necessary to conduct in-depth exploration of its mechanism. In addition, does the NOS-bridge affect the octameric structure of HTAncLAAO2 or the substrate preference to different amino acids?
3. The introduction of related researches on enzymatic synthesis of D-AAAs is insufficient.

Moreover, what are the advantages and disadvantages of the enzyme-catalyzed production of D-Trp in this study compared to the reported enzyme-catalyzed production of D-AAAs?

4. The study fails to address how the findings are specific and important for inadequate results and discussion.

5. Some minor issues are listed as follows:

(1)L33: 'shows >6-fold increase'  'shows a >6-fold increase'.

(2)L132: 'NOS-bride'  'NOS-bridge'.

Reviewer #3 (Remarks to the Author):

The work has been carefully described, the hypotheses are sound, the obtained results are well presented and the conclusions are reasonable. Therefore, I have no objections to publishing the work once a couple of issues have been addressed.

In particular, I want to start with the HPLC traces, as shown in Figure 6. Figure 6B shows only D and L forms, whereas, in the other two analogous figures, the racemate has been correctly shown. Please substitute L-1a with DL-1a (which can be obtained by simply mixing the two enantiomers, if not available).

In addition, the shown peaks are a bit too broad to justify ee>99% with a high degree of certainty, especially when another small peak (Is it really a peak? Has it been identified?) is always present just after the D-peak. From my experience, with that columns, it may be advantageous to add a small amount (5%-10%) of MeOH to the eluent to decrease tailing and sharpen the peaks.

Finally, a minor issue in Figure 5: it is a bit confusing to have on the two vertical axes two percentage values differently scaled. Please consider simply displaying the product e.e. in dependence on time in order to improve readability without losing any quality of the data. So much for science. Moving to other matters, I am inclined to suggest a careful proofreading of the manuscript by a native English speaker.

Answer to reviewer 1

We deeply appreciate the reviewer both for his/ her patience, valuable time, and precise comments/ questions. Our answers were written in the followings. Modification points were highlighted in red color in the manuscript. The comments/ questions are shown in italic.

1. It is claimed that initial assessment of HTAncLAAO2's specific activity is similar to the HTAncLAAO (Fig. 2A). The specific activity of HTAncLAAO should be given in Fig 2A.

Ans. Accordingly, we added the data for specific activity of HTAncLAAO toward 20 L-AAs in the Fig. 2A. (Please check the Fig 2A in the manuscript)

2. The NOS bridge identified in HTAncLAAO2 had minimal influence on in vitro parameters such as enzyme activity and thermostability. It is unimportant and should be simplified.

Ans. We apologize for not following to this comment. Reviewer 2 recommended a comprehensive analysis of the NOS bridge observed in the HTAncLAAO2 structure (refer to comment 2 by Reviewer 2). It is crucial to highlight that the NOS bridge does not consistently affect enzyme activity because there is no research reporting that the NOS bridge may be incidentally formed by reaction products of oxidases. To complement this point, we have incorporated the following sentences into the manuscript.

(After) **HTAncLAAO2 generates H₂O₂ during the reaction, potentially creating NOS bridge between K304 and C505, which are located in close proximity. These findings indicate that the NOS bridge does not always influence enzyme activity, suggesting the necessity for experimental analysis to evaluate the properties of this bridge. There is also possibility that the bridge might have other functions that cannot be characterized by in vitro assays. (line 164–169, P6)**

3. The authors attempted to create an HTAncLAAO2 variant with high activity towards L-Trp. Why L-Trp? How about other L-AAs that are not preferential substrates for HTAncLAAO2? At least it should be discussed. If more L-AAs were catalyzed by other HTAncLAAO2 variants guided by structural information, the manuscript will be more important.

(Ans) Accordingly, we measured relative activity toward 20 L-AAs for HTAncLAAO2(W220A) variant. As shown in Table S5, HTAncLAAO2(W220A) exhibited activity toward a long chain L-AAs (such as L-Trp, L-Lys and L-Arg). The broad substrate selectivity of the W220A was brought by introducing a cavity at the active site by compromising affinity (Km values) for L-Phe, L-Leu, L-Ile and L-Met. To complement this, we added the following sentences in the manuscript.

(After) The relative activity of the HTAncLAAO2(W220A) variant towards 20 L-AAs was evaluated, showing a selectivity conversion because of the mutation; this variant acquired the activity toward long-chain L-AAs (L-Trp, L-Lys and L-Arg) by sacrificing with the activity toward L-Ile and L-Val (Table S5). Enzyme kinetic analysis suggested that the W220A mutation likely introduces a cavity in the active site, accommodating the long-chain L-AAs while compromising affinity for L-Phe, L-Leu, L-Met, and L-Ile. Indeed, the K_m values of HTAncLAAO2(native) for these four L-AAs were at least 3-fold lower than those of the HTAncLAAO2(W220A) variant (Table S6). (line 202–209, P7)

In this study, we sought to develop a reaction system for synthesizing D-Trp derivatives, anticipating their potential as bioactive compounds for future medicinal applications. To emphasize this point, we have incorporated the subsequent sentence.

(After) D-Trp derivatives act as inhibitors of enteric pathogens³² and play a role in bioactive secondary metabolites³³. Establishing a reaction system to synthesize these derivatives can advance research in these areas by ensuring a consistent supply of the compounds. (line 95–97, P4)

Currently, we cannot obtain other HTAncLAAO2 variants that exhibit activity toward other L-AAs because many of the variants were insolubilized. We are now proceeding to obtain new variants that have novel activity.

Answer to reviewer 2

We deeply appreciate the reviewer both for his/ her patience, valuable time, and precise comments/ questions. Our answers were written in the followings. Modification points were highlighted in red color in the manuscript. The comments/ questions are shown in italic.

1. The activity detection of the control group (AncLAAO) in Fig.2D at 30℃ should be set at a smaller time interval. From the current experimental results, it cannot be seen that the activity of AncLAAO decreases over time.

We have added data on the relative activity of AncLAAO, sampled at smaller time intervals. Please refer to Figure 2A in the main text.

2. Reported studies showed that NOS-bridge can affect the structure, function, and stability of proteins. However, the results of this study show that NOS-bridge only has a certain effect on the activity of HTAncLAAO2, but has no significant effect on its thermal stability. What is the reason? In my opinion, It is necessary to conduct in-depth exploration of its mechanism. In addition, does the NOS-bridge affect the octameric structure of HTAncLAAO2 or the substrate preference to different amino acids?

To answer this comment, we assessed the oligomeric state and substrate specificity of HTAncLAAO2 and its variants, C505A and K304A. Both variants exhibited an 8-mer structure, consistent with HTAncLAAO2. Indeed, the elution times of HTAncLAAO2 (10.34 mL), C505A (10.39 mL), and K304A (10.40 mL) were nearly indistinguishable (Fig. S2). No remarkable differences in substrate specificity between HTAncLAAO2 and its variants were observed. Given these results, we predicted that the NOS bridge may have formed contingently due to the proximal locations of both C505 and K304. Further details have been incorporated into the manuscript.

(After)

a. Gel-filtration chromatography analysis of the variants indicated that the NOS bridge is not affected to form the oligomer state of HTAncLAAO2; the variants had 8-mer in solution condition as well as the native form (Fig. S2). (line 160–162, P6)

b. HTAncLAAO2 generates H₂O₂ during the reaction, potentially creating NOS bridge between K304 and C505, which are located in close proximity. These findings indicate that the NOS bridge does not always influence enzyme activity, suggesting the necessity for experimental analysis to evaluate the properties of this bridge. There is also possibility that the bridge might have other functions that cannot be characterized by *in vitro* assays. (line 164–169, P6)

3. The introduction of related researches on enzymatic synthesis of D-AAs is insufficient. Moreover, what are the advantages and disadvantages of the enzyme-catalyzed production of D-Trp in this study compared to the reported enzyme-catalyzed production of D-AAs?

Accordingly, we added sentences to describe enzyme synthesis of D-AAs as follows. In short, we added description about the enzyme synthesis of D-AAs utilizing hydantoins. (After) *Several of enzymatic synthesis methods currently exist, such as the sequential conversion of hydantoin to D-amino acids via hydantoin racemization and D-selective degradation, which involve the use of three enzymes: hydantoin racemase, D-hydantoinase, and D-carbamoylase¹⁴. The reductive amination of keto acids into D-AAs is commonly adopted due to its capacity to generate D-AAs in large quantities with high enantio-purity¹⁵; the reaction can be performed by D-amino acid dehydrogenase¹⁶, D-amino acid aminotransferase¹⁷ and ω -transaminase¹⁸. (line 55–61, P3)*

Additionally, we discussed the merits and demerits of D-AAs synthesis using LAAOs in comparison to alternative methods. Key advantages of LAAO-based synthesis include the ability to use purified enzymes and the sole requirement of O₂ molecules for FAD regeneration. However, a demerit is the production of H₂O₂, which can lead to undesirable side reactions. Further elaboration on these points has been included in the manuscript.

(After) *Both LAAOs and LAADs present distinct advantages and disadvantages for the synthesis. An advantage of LAAOs is that the enzyme only requires O₂ molecules to regenerate oxidized FAD. Thus, purified LAAOs can be utilized as biocatalysts. In contrast, LAADs necessitate electron carriers such as phenazine methosulfate or E. coli membranes for the regeneration²¹⁻²³. However, LAAOs have a disadvantage in that they produce H₂O₂ as a by-product, which often leads to unexpected reactions during synthesis. In contrast, LAADs do not generate H₂O₂, thereby mitigating the risk of unexpected reactions²¹. (line 62–69, P3)*

4. The study fails to address how the findings are specific and important for inadequate results and discussion.

In response to these feedbacks, we have modified the manuscript with the following clarifications:

a. We examined the substrate specificity of the HTAncLAAO2(W220A) variant. Our analysis demonstrates that this mutation enhances substrate selectivity.

(After) The relative activity of the HTAncLAAO2(W220A) variant towards 20 L-AAs was evaluated, showing a selectivity conversion because of the mutation; this variant acquired the activity toward long-chain L-AAs (L-Trp, L-Lys and L-Arg) by sacrificing with the activity toward L-Ile and L-Val (Table S5). Enzyme kinetic analysis suggested that the W220A mutation likely introduces a cavity in the active site, accommodating the long-chain L-AAs while compromising affinity for L-Phe, L-Leu, L-Met, and L-Ile. Indeed, the K_m values of HTAncLAAO2(native) for these four L-AAs were at least 3-fold lower than those of the HTAncLAAO2(W220A) variant (Table S6). (line 202–209, P7)

b. We added the research background of LAAOs. HTAncLAAO2 had the similar folding to other LAAOs, such as L-Lysine α -oxidase and LAAO from *Pseudoalteromonas* species, but the oligomeric state of HTAncLAAO2 is unique. To complement this point, we added the following sentences.

(After) The overall structure of HTAncLAAO2, determined at a 2.2 Å resolution (Table S4), indicating that the enzyme has typical folding of amine oxidase superfamily, such as L-Lysine α -oxidase^{34, 35} and LAAO from *Pseudoalteromonas* species^{26, 36}, in spite that they shared low sequence identity to each other. The unique point of HTAncLAAO2 is that the enzyme displays a distinctive octameric state (Fig. 1A). (line 110–114, P5)

5. Some minor issues are listed as follows:

(1) L33: 'shows >6-fold increase'  'shows a >6-fold increase'.

Accordingly, we amended the suggested point. (line 33, P2)

(2) L132: 'NOS-bridge'  'NOS-bridge'.

Accordingly, we amended the suggested point. (line 148, P6)

Answer to reviewer 3

We appreciate you to read our manuscript carefully and give kind comments. Our answers were written in followings. Modification points were highlighted by red color in the manuscript. The comments/ questions are shown as italic.

In particular, I want to start with the HPLC traces, as shown in Figure 6. Figure 6B shows only D and L forms, whereas, in the other two analogous figures, the racemate has been correctly shown. Please substitute L-1a with DL-1a (which can be obtained by simply mixing the two enantiomers, if not available).

Accordingly, we amended the Figure 6B. In detail, D,L-1a was purchased, and deracemization was performed by following the procedures written in Material & Methods section. (Please refer to the Fig 6B in the manuscript)

In addition, the shown peaks are a bit too broad to justify ee>99% with a high degree of certainty, especially when another small peak (Is it really a peak? Has it been identified?) is always present just after the D-peak. From my experience, with that columns, it may be advantageous to add a small amount (5%-10%) of MeOH to the eluent to decrease tailing and sharpen the peaks.

Thank you for the helpful comments. To follow this, HPLC analysis was performed utilizing running buffer containing both HClO₄ and 5–10% (v/v) Methanol. As pointed out, the small peak confirmed in previous study can be separated as shown in Fig. 6B–D. The retention time of the separated peak was shifted >0.5 min compared with a peak of L-forms (corresponding to >0.4 mL elution volume), suggesting that the peak appeared after the reaction may be derived from the reaction products. The L-isomers were completely disappeared during the reaction. To complement this point, we amended the manuscript as follows.

- a. The HPLC analysis was performed by changing the running buffer containing HClO₄ and CH₃OH. (Please refer to Fig. 6B–D)
- b. The detailed running conditions of HPLC were represented in Table S7.

Finally, a minor issue in Figure 5: it is a bit confusing to have on the two vertical axes two percentage values differently scaled. Please consider simply displaying the product e.e. in dependence on time in order to improve readability without losing any quality of the data.

Accordingly, we amended the Figure 5. In short, we removed the plots about conversion of D- and L-form. (Please refer to Fig. 5B)

So much for science. Moving to other matters, I am inclined to suggest a careful proofreading of the manuscript by a native English speaker.

Accordingly, our manuscript was checked by native English speaker.

REVIEWERS' COMMENTS:

Reviewer #1 (Remarks to the Author):

After revision, it can be accepted for publication.

Reviewer #2 (Remarks to the Author):

After revision, the manuscript could be accepted by this journal.

Reviewer #3 (Remarks to the Author):

Very shortly put, the authors have conveniently addressed all the questions raised by the referees, so I believe the manuscript is now ready for publication.